# Sesquiterpenes from *Artemisia annua* and Their Cytotoxic Activities

**DOI:** 10.3390/molecules27165079

**Published:** 2022-08-10

**Authors:** Xiao Han, Yao Chai, Cheng Lv, Qianqian Chen, Jinling Liu, Yongli Wang, Guixin Chou

**Affiliations:** 1The MOE Key Laboratory of Standardization of Chinese Medicines and SATCM Key Laboratory of New Resources and Quality Evaluation of Chinese Medicines, Institute of Chinese Materia Medica (ICMM), Shanghai University of Traditional Chinese Medicine (SHUTCM), Shanghai 201203, China; 2Shanghai University(SHU), Shanghai 200444, China; 3Shanghai R & D Center for Standardization of Chinese Medicines, Shanghai 201203, China

**Keywords:** *Artemisia annua*, Asteraceae, sesquiterpenes, cytotoxic

## Abstract

*Artemisia annua* is a well-known traditional Chinese medicine. Due to its highest antimalarial efficacy, China has a long history of cultivating *A. annua*, and it is used for “clearing heat and detoxicating”. Several, studies have shown that the *A. annua* extract exerts cytotoxicity. In order to clarify the basis of the cytotoxic effect of *A. annua*, 18 sesquiterpenes were isolated from the herb, including 2 new sesquiterpenes and 16 known analogues. The structures of new compounds were elucidated by comprehensive spectroscopic analyses, including HR-ESI-MS, NMR experiments, single-crystal X-ray, and DP4+ and electronic circular dichroism (ECD) calculations. Cytotoxic activity screening revealed three compounds that exhibited cytotoxicity in a dose-dependent manner. Additional exploration showed that compound **5** significantly inhibited the proliferation of CT26 and HCT116 cells and induced apoptosis of HCT116 cells after 24 h. These chemical constituents contributed to elucidating the mechanism of action of the cytotoxic activity of *A. annua*.

## 1. Introduction

*Artemisia annua* L. (Asteraceae), also known as Qing Hao, is widely distributed in the temperate, cold-temperate, and subtropical regions of Europe and Asia [1]. In 168 BC, *A. annua* was used to treat diseases (52 prescriptions) in China [2]. In 340 AD, the Handbook of Prescriptions for Emergency Treatment (Zhou Hou Bei Ji Fang) first recorded the antimalarial effects of *A. annua* [3]. Currently, as the most famous ingredient in traditional Chinese medicine [4], artemisinin is the most effective antimalarial drug recognized by the World Health Organization (WHO). Traditionally, aerial parts of *A. annua* are collected and made into teas or decoctions, which are documented as exerting positive effects on reducing fever and preventing malaria, “clearing heat and detoxicating” [5]. Due to its strong flavor (rich in volatile oil) [6], *A. annua* is also used in perfumes and as a flavoring agent.

Natural products are the treasure house for discovering new medicines. Due to its highest efficiency in treating malaria with minimal side effects, *A. annua* has gained increasing attention. Several, researchers have found that the *A. annua* extract has significant cytotoxicity against tumor cells [7,8,9,10], especially artemisinin and its derivatives [11,12]. Many studies projects have been focused on developing new anticancer drugs from *A. annua*. Reportedly, artemisinin and its derivatives showed adequate cytotoxic activity from nano to micro range [13]. The endoperoxide contained in artemisinin is closely related to its cytotoxic activity [14]. Some studies determined the cytotoxicity of artemisinin and its derivatives, indicating significant efficacy of the compound on various tumor cells, such as lung cancer [15], breast cancer [16], ovarian cancer [17], endometrial cancer [18], and stomach cancer [19].

In order to further identify compounds with good cytotoxic biological activity to clarify the material basis of cytotoxicity in *A annua*, 18 sesquiterpenes of various structural types, including 2 new and 16 known compounds, classified into cadinene (**1**–**3**, **4**–**9**, **13**–**17**), caryolane (**10**), clovane (**11**), and eudesmane (**12**, **18**), have been isolated and identified in this plant. Notably, compounds **1**, **2**, **6**, and **14** were five-membered lactones tricyclic sesquiterpenes, compounds **7**, **16**, and **17** were six-membered lactones tricyclic sesquiterpenes, and compound **12** possesses an uncommon 6,10-ether bridged tricyclic framework. Compounds **10**, **12**, **13**, and **18** were isolated from *A annua* for the first time. Compounds **2**, **5**, and **16** inhibited colon cancer cell lines (HCT116 and CT26). The IC_50_ values of these compounds against HCT116 were 20.0, 16.7, and 21.5 μM, respectively, better than the positive control (CPT11, 24.4 μM). Furthermore, mechanistic studies showed that compound **5** inhibits the proliferation of CT26 cells. Moreover, compound **5** can induce apoptosis and inhibit the proliferation of HCT116 cells by blocking their G1 phase.

## 2. Results and Discussion

The dry aerial part of *A. annua* was boiled in water and extracted by ethyl acetate (EtOAc) and n-butanol to afford three residues, which were purified by column chromatography (CC) on silica gel or Sephadex LH-20, to yield two new natural compounds. The NMR spectra of compounds **1** and **2** and the DP4+ data of tested compounds are available as Appendix A.

In addition to the above new compounds, 16 known sesquiterpenoids (Figure 1) were also obtained: artemanin B (**3**) [20], (1*S*, 6*R*, 7*R*, 10*R*)- 6-carboxy-10-methyl-α-methylene-1-(1-oxobutyl)-cyclohexaneacetic acid (**4**) [21], artemisinic acid (**5**) [22], arteannuin B (**6**) [23], deoxyartemisinin (**7**) [24], 6,7-dehydroartemisinic acid (**8**) [25], arteannoide I (**9**) [26], caryolane-1,9β-diol (**10**) [27], (-)-clovane-2,9-diol (**11**) [28], 6, 15α-epoxy-1β, 4β-dihydroxy eudesmane (**12**) [29], pulioplopanones A (**13**) [30], qinghaosu I (**14**) [31], (1*R*, 4*R*, 4a*S*, 8*S*, 8a*S*)-decahydro-8-hydroxy-4-methyl-7-methylene-1-naphthalenecarboxylic acid (**15**) [31], annulide (**16**) [32], isoannulide (**17**) [32], and cryptomeridiol (**18**) [33].

The molecular formula of compound **1** was C_15_H_20_O_3_ based on the HR-ESI-MS (*m/z* 249.1481 [M + H]^+^, calculated for 249.1485). The IR spectrum of compound **1** indicated characteristic absorptions for hydroxyl at 3437 cm^−1^, lactone carbonyl at 1745 cm^−1^, and a terminal double bond at 1661 cm^−1^. The ^1^H NMR spectrum of **1** (Table 1) revealed characteristic signals of two terminal olefinic methylenes at *δ*_H_ 5.67 (1H, d, *J* = 1.3 Hz, H-13a), 6.09 (1H, d, *J* =1.5 Hz, H-13b), 4.79 (1H, t, *J* = 1.9 Hz, H-15a), and 4.86 (1H, t, *J* =2.1 Hz, H-15b), and one methyl at *δ*_H_ 0.97 (3H, d, *J* = 6.6 Hz, H-14). The ^13^C NMR and DEPT spectra exhibited 15 carbon resonances that were assigned to one methyl (*δ*_C_ 20.4), six methylenes (*δ*_C_ 26.7, 30.4, 30.1, and 32.1, including two olefinics, *δ*_C_ 121.6 and 113.0), four methines (*δ*_C_ 41.3, 42.4, and 31.1, including an oxygenated *δ*_C_ 76.9), and four quaternary carbons (a carbonyl *δ*_C_ 172.0, an oxygenated *δ*_C_ 87.9, and two olefinics, *δ*_C_ 148.0 and 143.7). These spectroscopic features were similar to the known compound arteannuin L [34]. The difference between these compounds was the presence of an exocyclic double bond group in **1**, instead of a methyl group at C-11 in arteannuin L, which was confirmed by HMBC correlations (Figure 2) of H_2_-13/C-7/C-11 and C-12.

The NOESY correlations (Figure 2) of H-1/H_3_-14 revealed that H-1 and H_3_-14 were cofacial and randomly assigned as *β*-oriented, and the correlation of H-5/H-7 showed they were on the opposite sides and assigned as *α*-oriented, while the relative configuration of C-6 was determined as *S**, by quantum-chemical ^13^C NMR calculations based on DP4+ probability analysis (Figure 3). The absolute configuration of **1** was established as 1*R*, 5*S*, 6*S*, 7*S*, and 10*S* by ECD experiment and calculation (Figure 4). Consequently, the structure of **1** was elucidated and named arteannuin P.

Compound **2** was obtained as yellow oil, with the molecular formula of C_15_H_20_O_3_ based on its HRESIMS (*m/z* 249.1488 [M + H]^+^, calculated for 249.1485), and its molecular formula was C_15_H_20_O_3_. A careful analysis of the ^1^H and ^13^C-NMR data revealed that **2** (Table 1) had a structural skeleton analogous to **1,** the major difference being an exomethylene group (δH 4.86, 4.79, δC 113.0) that was present in **1** instead of the olefinic methane (δH 5.44, δC 121.6) in **2**. This finding indicated that the exocyclic double bond Δ^4–15^ in **1** was replaced by an endocyclic double bond Δ^4–5^ in **2**, which was further confirmed by HMBC correlations from H_3_-15 to C-3, C-4, and C-5. Based on the 2D NMR spectra and the chemical shift of C-7 (from *δ*_C_ 42.4 in **1** to *δ*_C_ 77.2 in **2**), the hydroxyl group should be connected to C-7 directly. The NOESY spectrum of **2** exhibited a correlation between H-1 and H_3_-14, suggesting these were on the α-face. As C-6 and C-7 were quaternary carbons without protons, the relative configuration of C-6 and C-7 could not be determined by the NOESY experiment. Thus, ^13^C NMR data of four different stereoisomers (6*R**,7*R**-**2**, 6*R**,7S*-**2**, 6*S**,7*R**-**2**, and 6*S**,7*S**-**2**) were calculated by using the GIAO method at the mPW1PW91/6-31+G** level in PCM methanol using Gaussian 09. Subsequently, using relative free energies at the wB97M-V/def2-TZVP level in SMD methanol, using ORCA Shielding constants, were used to calculate DP4+ probability analysis. Based on the evaluation result, 6*R** and 7*R** (100%) were the best fitness stereoisomers (Figure 3). The absolute configuration of **2** was assigned as 1*S*, 6*R*, 7*R*, and 10*R* according to the ECD calculation. Consequently, the structure of **2** was elucidated and named arteannuin Q.

Compound **3** [18] is a known compound, but the crystal structure has been reported. Herein, we supplemented its crystal structure. Compound **3** was obtained as a colorless needle crystal (CCDC: 2110137), and its absolute configuration was established as 1*S*, 3*R*, 4*R*, 5*R*, 6*R*, and 9*R* by X-ray crystallography using Cu Kα radiation.

The cytotoxicity of all isolated sesquiterpenoid compounds from *A. annua*, except compound **8** (too little), were evaluated in HCT116 and CT26 cells and compared to the positive control (CPT11). Compounds **2**, **5**, and **16** exhibited good cytotoxic activity against the HCT116 cell line with IC_50_ values (Table 2) of 20.0, 16.7, and 21.5 μM, which were better than those of CPT11 (24.4 μM). The most active compound **5** was selected for further research. EdU experiments (Figure 5) showed that compound **5** inhibits the proliferation of CT26 cells, which was confirmed by Western blot results (Figure 6). In HCT116 cells, Bcl2, p-CDK1, Cyclin E, and Cyclin D were downregulated, and P21, Bax, Cleaved caspase 3, and other genes (Figure 7) and proteins were upregulated, indicating that compound **5** not only inhibits HCT116 cells’ proliferation but also induces apoptosis. It may cause cell death by blocking the G1 phase.

Compounds **1** and **2** were isomers with different positions of alkenyl and hydroxyl. While **2** had a good activity of inhibiting colon cancer cell proliferation, the activity of **1** was not obvious. Moreover, the absolute configuration of the two compounds was markedly different. Therefore, the difference in the activities of **1** and **2** could be attributed to the differences in their 3D configurations.

## 3. Materials and Methods

### 3.1. General Experimental Procedures

The Anton Paar MCP5500 polarimeter (Anton Paar GmbH, Graz, Austria) was used to determine the optical rotations. ECD data were recorded on a Chirascan spectropolarimeter (Applied Photophysics Ltd., Surrey, UK). IR spectra were determined by a PerkinElmer FT-IR spectrometer (PerkinElmer Inc, Waltham, Massachusetts, USA). The NMR experiments were recorded using Bruker AVANCE-III instruments (Bruker Co. Ltd., Bremen, German). HRESIMS data were determined with a Waters UPLC Premier QTOF ((Waters, Manchester, U.K). The crystal data were selected on a Bruker APEX-II CCD diffractometer (Bruker Co. Ltd., Bremen, German). Agilent 1260 systems (Agilent Technologies, Santa Clara, CA, USA) were used for HPLC that was carried out on a Shiseido Capcellpak C18 (250 × 20 mm). CC was carried out with ODS C18 (45–60 μm, YMC Co., Ltd., Kyoto, Japan), silica gel (100–200 and 300–400 mesh, Qingdao Haiyang Chemical, Qingdao, China), and Sephadex LH-20 (Amersham Biosciences, Amersham, UK). TLC and preparative TLC were performed with HSGF254 silica gel plates (Yantai Jiangyou Silica Gel Development Co., Ltd., Yantai, China). RPMI 1640 cell medium, McCoy’s 5A cell medium, FBS, and antibiotics (all from Thermo Fisher Scientific, Waltham, MA, USA) were for cell culture. Cell Counting Kit-8 (Shanghai YEASEN Biotechnology Co., Ltd., Shanghai, China) was used to determine cell viability. The EdU Cell Proliferation kit (Guangzhou Ribo Biotechnology Co., Ltd., Guangzhou, China) was used to test the cell proliferation activity. TRIzol reagent (Thermo Fisher Scientific, Waltham, MA, USA) was used to extract total RNAs, the cell lysis buffer (Dalian Meilun Biotechnology Co., Ltd., Dalian, China) was used to extract total protein, the BCA protein assay kit (Shanghai YEASEN Biotechnology Co., Ltd., Shanghai, China) was used to determine the protein content, and Super ECL Detection Reagent (Shanghai YEASEN Biotechnology Co., Ltd., Shanghai, China) was used for the visualization of protein bands.

### 3.2. Plant Material

The plants were obtained from Chongming Island (Shanghai, China), and the plant was identified by Prof. Gui-Xin Chou (Shanghai University of Traditional Chinese Medicine). The specimens were deposited in the Research Institute of Chinese Medicine, Shanghai University of Traditional Chinese Medicine Institute (specimen no. 20190920-1).

### 3.3. Extraction and Isolation

The dry aerial part of *A. annua* (32.5 kg) were sliced into 0.5–1.0 cm pieces and boiled three times in distilled water (325 L, 260 L, and 260 L, respectively). The pooled filtrates were concentrated to a density of 1.15–1.20 to obtain residues, which were then were extracted by EtOAc and n-butanol three times, respectively. The EtOAc portion (273.35 g) was subjected to CC on silica gel (100–200 mesh) using petroleum ether-CH_2_Cl_2_ (5:1 to 0:1) and CH_2_Cl_2_-MeOH (200:1 to 10:1) to collect 14 fractions (Fr. 1–14). Fr. 13 (15.48 g) was separated by ODS CC and eluted with MeOH-H_2_O (30%–100%) to obtain 42 fractions (Fr. 13–1 to Fr. 13–42). Fr. 13–23 were subsequently loaded on Sephadex LH-20 column with MeOH (100%) solvent to obtain 9 fractions (Fr. 13–23–1 to Fr. 13–23–9). Compound **3** (12.5 mg) was isolated from Fr. 13–23–3 (185.6 mg) by preparative HPLC (ACN-H_2_O, 46%). Compound **4** (5.3 mg) was yielded from Fr. 13–23–7 by preparative TLC. Fr. 13–29 was separated on the Sephadex LH-20 column with MeOH-H_2_O (80%) solvent to obtain two fractions (Fr. 13–29–1, Fr. 13–29–2) that were purified by preparative TLC to yield compound **9** (2.8 mg) and compound **10** (14.9 mg). Fr. 13–18 and Fr. 13–31 were separated by a similar procedure to yield compounds **11** (13.7 mg), **18** (2.1 mg), and compound **12** (5.2 mg). Fr. 5 (8.3 g) and Fr. 6 (5.3 g) were subjected to CC on silica gel (100–200 mesh) using petroleum ether-CH_2_Cl_2_ (10:1 to 1:5) to obtain 22 and 20 fractions, respectively (Fr. 5–1 to Fr. 5–22 and Fr. 6–1 to Fr. 6–20). Compound **5** (58.5 mg) was repeatedly recrystallized with petroleum ether from Fr. 5–2. Then, Fr. 5–7 was separated by Sephadex LH-20 column and eluted with petroleum ether: CH_2_Cl_2_: MeOH = 5:5:1 to obtain 7 fractions (Fr. 5–7–1 to Fr. 5–7–7). Fr. 5–7–2 was isolated by preparative TLC to yield compound **7** (20.1 mg). Fr. 5-7-3 used the same protocols as Fr. 5–7–2 and was further purified by preparative HPLC (ACN-H_2_O, 58%) to retrieve compound **15** (6.3 mg), compound **16** (12.8 mg), and compound **17** (2.3 mg). Compound **6** (19.0 mg), compound **13** (10.4 mg), and compound **14** (13.3 mg) were separated from Fr. 5–14 and Fr. 5–8 using the same separation methods as Fr. 5–7. Then, Fr. 6–15 and Fr. 6–8 were subjected to Sephadex LH-20 column and eluted with petroleum ether: CH_2_Cl_2_: MeOH = 5:5:1 and further purified by preparative HPLC (ACN-H_2_O 53%, 53%, and 88%, respectively) to obtain compound **1** (7.5 mg), compound **2** (3.3 mg), and compound **8** (0.7 mg).

*Compound* **1**: yellowish oil; [*α*]^25^_D_ − 55 (*c* 0.1, in MeOH); IR (KBr) Vmax: 3437, 2926, 1745, 1661, 1406, 1262, 1156, and 955 cm^−1^; ^1^H and ^13^C NMR data (Table 1); HR-ESI-MS *m*/*z* 249.1481 [M + H]^+^ (calculated for 249.1485).

*Compound* **2**: yellowish oil; [α]^25^_D_ + 138 (*c* 0.1, in MeOH); IR (KBr) Vmax: 3435, 2929, 1739, 1672, 1378, 1275, 1192, and 911 cm^−1^; ^1^H and ^13^C NMR data, see Table 1; HR-ESI-MS *m/z* 249.1488 [M + H]^+^ (calculated for 249.1485).

*Compound* **3**: colorless, needle-like crystals; [α]^25^_D_ − 37 (*c* 0.1, in MeOH); crystal data: *a* = 16.0730(5) Å, *b* = 5.3794(2) Å, *c* =18.3897(6) Å, *α* = 90°, *β* =112.6250(10) °, *γ* = 90°, V = 1467.67(9) Å3, *μ*(Cu Kα) = 0.703 mm^−1^, Flack parameter = −0.08(11).

### 3.4. Quantum Chemical NMR and ECD Calculations of Compound **1**–**2**

Firstly, we used a random search in the Sybyl-X 2.0 using the MMFF94S force field with an energy cutoff of 2.5 kcal/mol to analyze the conformation; the lowest energy conformer was detected in compounds **1** and **2**. Subsequently, the NMR calculations were obtained by using the GIAO method at the mPW1PW91/6-31+G** level in PCM methanol using Gaussian 09. The conformational analysis of the calculated ECD used the TDDFT methodology at the PBE0/def2-TZVP level in MeOH and were simulated by the overlapping Gaussian function (half the bandwidth at ^1^/e peak height, sigma = 0.30 for all). The simulated spectra of the conformers were averaged according to the Boltzmann distribution theory and their relative Gibbs free energy (ΔG) to obtain the final spectra.

### 3.5. Cell Culture

CT26 (mouse colon cancer cell line) and HCT116 (human epithelial fibrous colon carcinoma) cell line obtained from Dr. Wei Dou (Shanghai University of Traditional Chinese Medicine) were cultured in 1640 medium and McCoy’s 5A medium, respectively, supplemented with 10% FBS and antibiotics (100 U/mL penicillin and 100 μg/mL streptomycin) at 37 °C under, 5% CO_2_. The cells were passaged every 2 days, and used for the following tests.

### 3.6. Determination of Cell Viability by CCK-8 Assay

Cell viability was assessed using the CCK-8 assay. The cells were seeded in a 96-well plate at a density of 1 × 10^4^ cells/well, cultured for 12–24 h to attach, and treated with different concentrations of the compounds for an additional 24 h. Next, 10% CCK-8 reagent was added to each well and incubated for an additional 0.5 h, and the absorbance was at 450 nm on the microplate reader. The data were analyzed using Prism 7.0 software (Graph Pad, Avenida, CA, USA).

### 3.7. Quantitative Real-Time Polymerase Chain Reaction (q-PCR) Analysis

The cells were seeded in a 6-well plate at a density of 2 × 10^5^ cells/well and treated with compound **5** (10, 20, and 30 μM, respectively). After 24 h of culture, total RNAs were extracted using TRIzol reagent, and *β*-actin served as an internal reference gene. The primers used in this experiment are listed in Table 3.

### 3.8. EdU Staining Assay

CT26 cells were seeded in a 96-well plate at 8 × 10^3^ cells/well, incubated at 37 °C in a 5% CO_2_ incubator for 24 h, and treated for an additional 24 h with media containing 10, 20, and 30 μM of compound **5**, respectively. Three replicate wells were set up for each concentration. Subsequently, EdU staining was performed in accordance with the instructions of the manufacturer. Then, cells were fixed, followed by Apollo staining and Hoechst staining, and then observed and photographed under an inverted fluorescent microscope.

### 3.9. Western Blotting Assay

An equivalent of 1 × 10^5^ cells/mL was seeded in 6-well plates in a volume of 2 mL, incubated for 24 h, and then treated for an additional 24 h with media containing 10, 20, and 30 μM of compound **5**, respectively. The total protein was extracted by lysis buffer containing protease inhibitors. Then, the protein content was determined by the BCA method, and samples of equivalent concentration were prepared. The samples from each group (25 μg/20 μL) were separated by 12% sodium dodecyl sulfate-polyacrylamide gel electrophoresis (SDS-PAGE) and transferred to polyvinylidene fluoride (PVDF) membranes. Then, the membrane was blocked with 5% nonfat milk at room temperature for 1 h, probed with the primary antibody at 4 °C overnight, and then incubated with the secondary antibody. Finally, the immunoreactive bands were detected by ECL.

## 4. Conclusions

In summary, 18 sesquiterpenoids were separated and identified from the ethyl acetate soluble partition of *A. annua*, including 2 new five-membered lactones tricyclic sesquiterpenes, and 4 of them were isolated from *A annua* for the first time. Among these, compounds **2**, **5**, and **16** showed adequate cytotoxicity on human and murine colon cancer cells. Interestingly, the mechanism of action of compound **5** on the two cells was different. In addition, compound **5** caused cell death by inhibiting the proliferation of the CT26 cells, which inhibited the proliferation as well as induced apoptosis in HCT116 cells. Our research not only enriched new active sesquiterpenes of *A. annua* with potential cytotoxicity, but also provided ideas to explore novel anti-tumor agents.

## Figures and Tables

**Figure 1 molecules-27-05079-f001:**
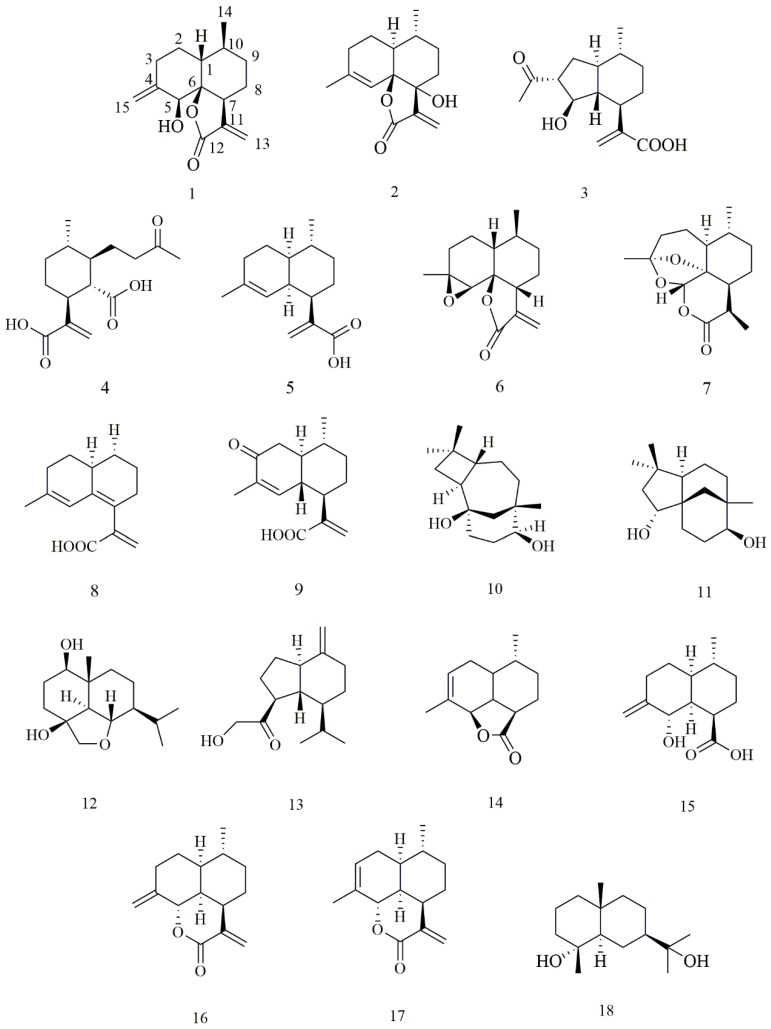
Chemical structures of **1**–**18** from *A. annua*.

**Figure 2 molecules-27-05079-f002:**
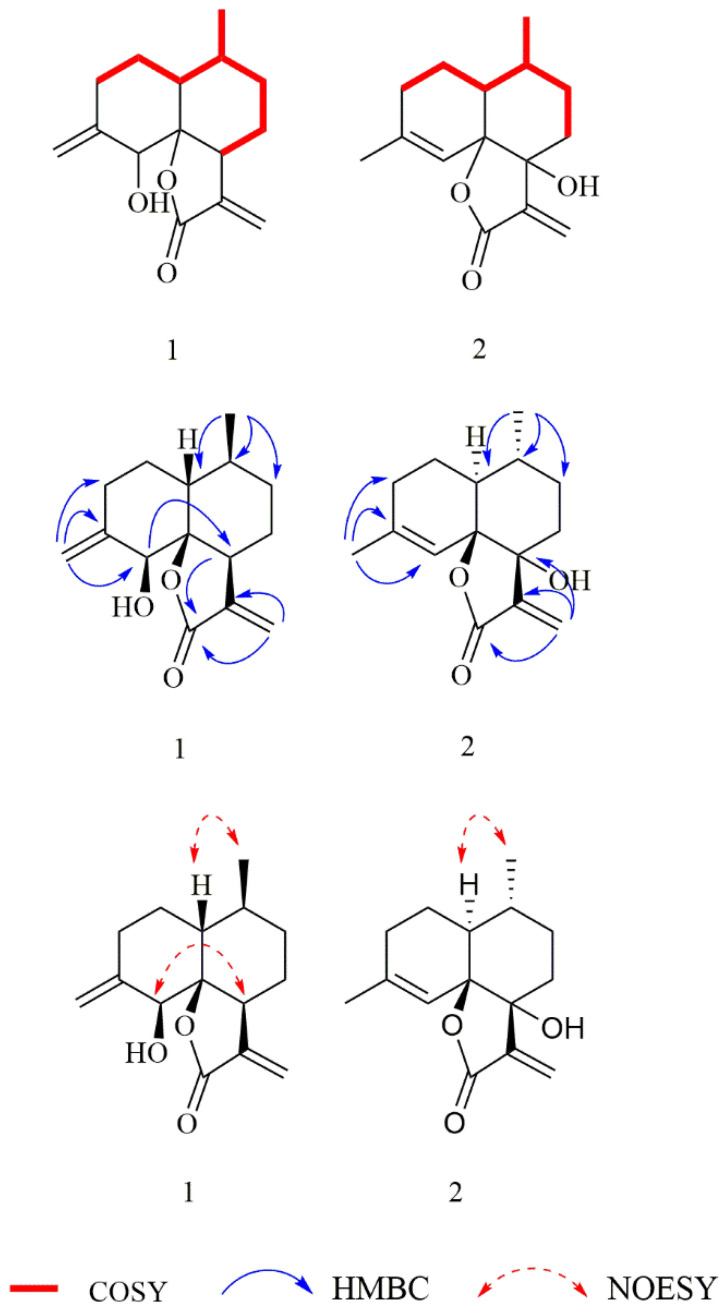
^1^H-^1^H COSY, HMBC, and ^1^H-^1^H NOESY correlations for **1**–**2**.

**Figure 3 molecules-27-05079-f003:**
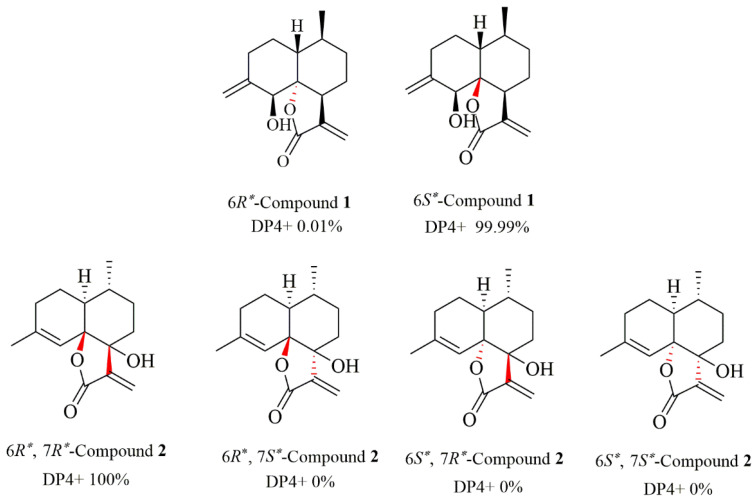
Results of DP4+ analysis for **1** and **2**.

**Figure 4 molecules-27-05079-f004:**
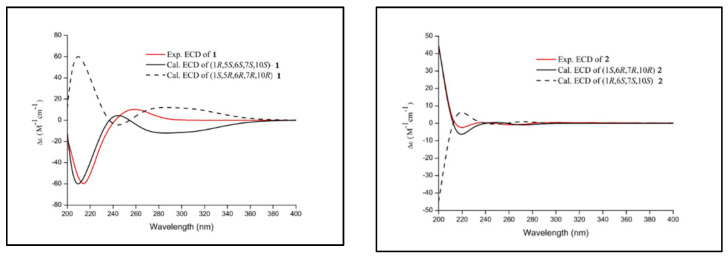
Experimental ECD spectra for compounds **1** and **2**.

**Figure 5 molecules-27-05079-f005:**
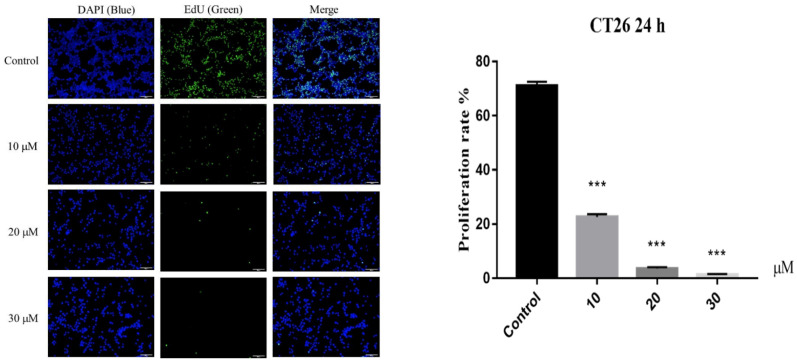
Cell proliferation assay of CT26 cell line at different concentrations of compound **5** (10, 20, and 30 μM) after 24 h. Data are present as the mean ± SD, compared with the control group. *** *p* < 0.001 (*n* = 3).

**Figure 6 molecules-27-05079-f006:**
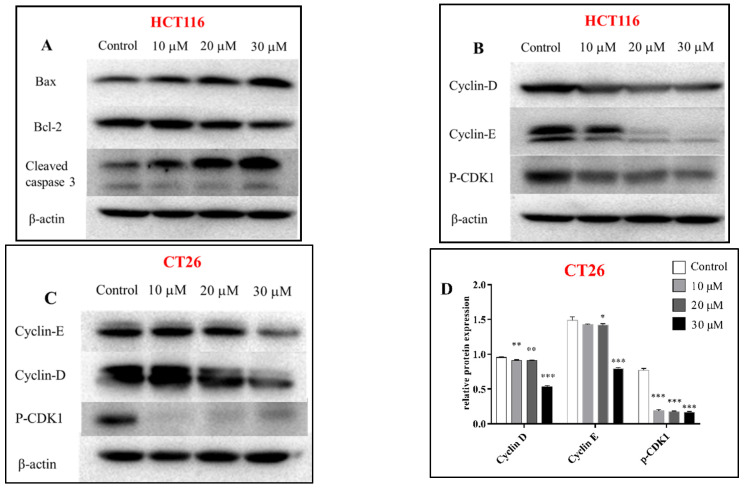
Western blot was used to analyze the expression of apoptosis and proliferation-related proteins in HCT116 (**A**,**B**) and CT26 (**C**) cells treated with compound **5** for 24 h; relative expression of proliferation proteins in CT26 cells (**D**), relative expression of apoptosis and proliferation proteins in HCT116 cells (**E**); *β*-actin was the internal reference. Data are present as the mean ± SD, compared with the control group. * *p* < 0.05, ** *p* < 0.01, *** *p* < 0.001 (*n* = 3).

**Figure 7 molecules-27-05079-f007:**
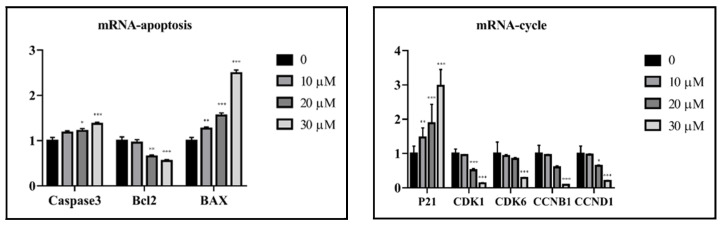
Compound **5** affected the gene expression of proliferative and apoptosis factors in HCT116 cells. Data are present as the mean ± SD, compared with the control group. * *p* < 0.05, ** *p* < 0.01, *** *p* < 0.001 (*n* = 3).

**Table 1 molecules-27-05079-t001:** ^1^H-NMR (400MHz) and ^13^C-NMR (100MHz) data of compounds **1** and **2** (MeOD, *δ* in ppm, *J* values in Hz).

NO	1		2	
	*δ*_C_ Type	*δ*_H_ (*J* in Hz)	*δ*_C_ Type	*δ*_H_ (*J* in Hz)
1	41.3, CH	1.75 (m)	43.4, CH	1.47 (d, 2.7)
2	26.7, CH_2_	1.35 (d, 3.8), 1.88 (m)	22.5, CH_2_	1.56 (ddd, 8.4, 3.7, 2.1), 1.95 (m)
3	30.4, CH_2_	2.24 (m), 2.44 (tdt, 13.6, 5.3, 2.0)	31.7, CH_2_	2.10 (s), 2.11 (d, 3.0)
4	148.0, C		144.1, C	
5	76.9, CH	3.60 (s)	121.6, CH	5.44 (q, 1.6)
6	87.9, C		87.4, C	
7	42.4, CH	3.21 (ddd, 10.5, 5.0, 3.4)	77.2, C	
8	30.1, CH_2_	1.29 (m), 1.95 (m)	37.0, CH_2_	1.66 (m), 1.81 (m)
9	32.1, CH_2_	1.16 (m), 1.61 (m)	30.4, CH_2_	1.35 (dd, 7.6, 3.8), 1.52 (m)
10	31.1, CH	1.33 (m)	30.8, CH	1.50 (s)
11	143.7, C		147.6, C	
12	172.0, C		171.0, C	
13	121.6, CH_2_	5.67 (d, 1.3), 6.09 (d, 1.5)	119.9, CH_2_	5.70 (s), 6.12 (s)
14	20.4, CH_3_	0.97 (d, 6.6)	19.8, CH_3_	1.00 (d, 5.9)
15	113.0, CH_2_	4.79 (t, 1.9), 4.86 (t, 2.1)	23.5, CH_3_	1.71 (d, 1.3)

**Table 2 molecules-27-05079-t002:** Cytotoxicity of compounds (IC_50_ values in μM).

Compounds	HCT116	CT26
**2**	20.0	49.9
**5**	16.7	14.9
**16**	21.5	53.5
CPT11	24.4	44.3

IC50 values were presented as mean ± SD’ (*n* = 3).

**Table 3 molecules-27-05079-t003:** Primers for RT-q PCR.

Gene	Forward Primer	Reverse Primer
*CDK4*	TTTTGAGCATCCCAATGTTGTC	TCGACGAAACATCTCTTGATCT
*CDK1*	CACAAAACTACAGGTCAAGTGG	GAGAAATTTCCCGAATTGCAGT
*CDK6*	CGAACAGACAGAGAAACCAAAC	CTCGGTGTGAATGAAGAAAGTC
*CCND1*	GTCCTACTTCAAATGTGTGCAG	GGGATGGTCTCCTTCATCTTAG
*CCNB1*	GACTTTGCTTTTGTGACTGACA	CCCAGACCAAAGTTTAAAGCTC
*β-actin*	CAGCCTTCCTTCTTGGGTAT	TGGCATAGAGGTCTTACGG
*Caspase3*	CACATGAUGCTUCCGACUGA	ATGGTTCGGTTACTACGGTCA
*BCL2*	GACTTCGCCGAGATGTCCAG	GAACTCAAAGAAGGCCACAATC
*Bax*	CGAACTGGACAGTAACATGGAG	CAGTTTGCTGGCAAAGTAGAAA

## Data Availability

All the data related to this article is available at https://www.mdpi.com (accessed on 17 July 2022).

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
