# Peer review of "Sesquiterpenes from Artemisia annua and Their Cytotoxic Activities"

_molecules, 2022, doi:10.3390/molecules27165079_

Round 1
Reviewer 1 Report
In this work, two new sesquiterpenes along with sixteen known ones were isolated from Artemisia annua. Their structures were determined by the NMR spectral data and DP4+ and electronic circular dichroism (ECD) calculations. However, the structure elucidation looks not firm. And the cytotoxicity bioassay was conducted for these compounds, but the results seem that they needs to be rechecked again. I will recommend it after revision.
1. Please cite the reference ‘Tu, Y. Artemisinin—a gift from traditional Chinese medicine to the world (Nobel lecture). Angew. Chem. Int. Ed. 2016, 55, 10210-10226.’ in the first paragraph of Introduction.
2. Please give the numbers for the sixteen known compounds corresponding to their names on Page 2.
3. Please give more detailed analysis of the results of DP4+ quantum-chemical 13C NMR calculations in the manuscript.
4. Please add the protocols of DP4+ quantum-chemical 13C NMR calculations in the corresponding section of the manuscript.
5. Fig. 18S and Fig. 19S consisted of several tables and figures. Please name them individually.
6. Please add the experimental and calculated ECD spectra in the manuscript, not put in the Supplementary Information.
Question 1: Have the sixteen known sesquiterpenes been isolated from the plant Artemisia annua? These sixteen known compounds and the two new ones exhibited several different skeletons. What are structural relationships between them?
Question 2: The NOESY correlations of H-1/H3-14 and H-5/H-7 observed for compound 1 only revealed each group of hydrogens are cofacial, without any indicative spatial relationships for the above mentioned two groups of hydrogens. Did you find any NOESY cross-peaks of these two groups of hydrogens? As reported for the reference compound arteannuin L in Ref. [32], all of the two groups of hydrogens were cofacial, which was different from the relative configurations given in this manuscript. Please check compound 2, too.
Question 3: Which compound was not subjected to cytotoxicity bioassay? As described in the Extraction and isolation section, the mass of compound 7 is 20.1 mg, much more than other compounds, absolutely not ‘compound 7 (too little)’ as recorded on Page 6.
Question 4: ‘compound 5 not only inhibits HCT116 cells proliferation but also induces apoptosis…The mechanism of action on these two cell lines could ascribe to the large difference in their IC50 values.’ Was this consistent with that the cytotoxicity IC50 values of compound 5 against HCT116 (16.7 μM) was more than against CT26 (14.9 μM) as shown in Table 2?
Corrections:
1. Abstract: ‘Several, studies’ → ‘Several, studies’
‘16 known species’ → ‘16 known analogues’
‘Cytotoxic activity screening identified, three compounds cytotoxicity in a dose-dependent manner’ → ‘Cytotoxic activity screening revealed three compounds exhibited cytotoxicity in a dose-dependent manner’
2. Introduction: ‘exert positive effects, re-duce fever, and prevent malaria’ → ‘exerting positive effects on reducing fever and preventing malaria’
‘Several, researchers’ → ‘Several researchers’
3. Results and Discussion:
P2: ‘aerial part f A. annua’ → ‘aerial part of A. annua’
‘extracted in ethyl acetate (EtOAc) and n-butanol to obtain three residues purified by column chromatography’ → ‘extracted by ethyl acetate (EtOAc) and n-butanol to afford three residues, which were purified by column chromatography’
‘compounds 1–2’ → ‘compounds 1 and 2’
‘artemanins B’ → ‘artemanin B’
‘Arteannoide I’ → ‘arteannoide I’
‘Qinghaosu I’ → ‘qinghaosu I’
P6: The expression ‘a sodium adduct ion peak at m/z 249.1488 [M + H]+’ was wrong.
‘olefinic methane’ → ‘olefinic methine’
This sentence ‘except for the disappearance of the exomethylene (δH 4.86, 4.79, δC 113.0) in 1 and the pres-ence of olefinic methane (δH 5.44, δC 121.6) in 2, as well as the position of a hydroxyl group’ was obscure.
‘experimental values of 5 using the DP4+’ → ‘experimental values of 2 using the DP4+’
‘its absolute configuration 1 was established’ → ‘its absolute configuration was established’
P6: ‘of inhibited’ → ‘of inhibiting’
References: Please check and revise some references according to the Instructions for Authors.
Author Response
Dear Reviewer,
Thank you very much for your advice on our manuscript titled “Sesquiterpenes from Artemisia annua and their cytotoxic activities (molecules-1845823)”. We submitted the revised manuscript as suggested. We have addressed the comments raised by reviewers, and the amendments are highlighted in red in the revised manuscript. We hope that the revision is acceptable and we look forward to hearing from you soon.
- Please cite the reference ‘Tu, Y. Artemisinin—a gift from traditional Chinese medicine to the world (Nobel lecture). Chem. Int. Ed. 2016, 55, 10210-10226.’ in the first paragraph of Introduction.
A: Thank you very much for your suggestion, this is a very important reference, we added it in the first paragraph of Introduction. (Page 1, reference 4)
- Please give the numbers for the sixteen known compounds corresponding to their names on Page 2.
A: Thank you very much for your suggestion, we added numbers to known compounds. (Page 2)
- Please give more detailed analysis of the results of DP4+ quantum-chemical 13C NMR calculations in the manuscript.
A: We added more analysis of the results of DP4+ quantum-chemical 13C NMR calculations in the manuscript. (Page 6)
13C NMR data of four different stereoisomers (6R*,7R*-2, 6R*,7S*-2, 6S*,7R*-2 and 6S*,7S*-2) were calculated by using the GIAO method at the mPW1PW91/6-31+G** level in PCM methanol using Gaussian 09. Subsequently, using relative free energies at the wB97M-V/def2-TZVP level in SMD methanol using ORCA Shielding constants were used to calculate DP4+ probability analysis. Based on the evaluation result, 6R*, and 7R* (100%) was the best fitness stereoisomer (Fig. 3).
- Please add the protocols of DP4+ quantum-chemical 13C NMR calculations in the corresponding section of the manuscript.
A: the protocols of DP4+ quantum-chemical 13C NMR calculations were added in the 4.4. Quantum chemical NMR and ECD calculations of compound 1-2. (Page 10)
- 18S and Fig. 19S consisted of several tables and figures. Please name them individually.
A: Fig. 18S and Fig. 19S have been modified as suggested.
- Please add the experimental and calculated ECD spectra in the manuscript, not put in the Supplementary Information.
A: The ECD spectra was added in the manuscript. (Fig. 4)
Question 1: Have the sixteen known sesquiterpenes been isolated from the plant Artemisia annua? These sixteen known compounds and the two new ones exhibited several different skeletons. What are structural relationships between them?
A 1: Compounds 10, 12, 13, 18 were isolated from A annua for the first time. 18 sesquiterpenes were classified into cadinene (1-3, 4-9, 13-17), caryolane (10), clovane (11) and eudesmane (12, 18). Notably, compounds 1, 2, 6, and 14 were five-membered lactones tricyclic sesquiterpenes, compounds 7, 16, and 17 were six-membered lactones tricyclic sesquiterpenes, and compound 12 possesses uncommon 6,10-ether bridged tricyclic framework. (Page 2)
Question 2: The NOESY correlations of H-1/H3-14 and H-5/H-7 observed for compound 1 only revealed each group of hydrogens are cofacial, without any indicative spatial relationships for the above mentioned two groups of hydrogens. Did you find any NOESY cross-peaks of these two groups of hydrogens? As reported for the reference compound arteannuin L in Ref. [32], all of the two groups of hydrogens were cofacial, which was different from the relative configurations given in this manuscript. Please check compound 2, too.
A:In the enlarged NOESY spectrum of compound 1, the H-1 showed correlation with Me-14, the H-5 showed correlation with H-7, moreover the NOESY cross-peaks of Me-14 with the H-5 and H-7 were not observed. Therefore, H-1 and Me-14 were on the same face as β-oriented, whereas H-5 and H-7 were on the opposite face as α-orientation (see below the enlarged NOESY spectrum of compound 1).
Key NOESY correlations in compound 1:
In the enlarged NOESY spectrum of compound 2, the NOESY cross-peak of H-1 and Me-14 can be observed (see below the enlarged NOESY spectrum of compound 2), so H-1 and Me-14 were on the same face. According to the results of DP4+ and ECD analysis, H-1 and H-14 was α-orientation.
Key NOESY correlations in compound 2:
Question 3: Which compound was not subjected to cytotoxicity bioassay? As described in the Extraction and isolation section, the mass of compound 7 is 20.1 mg, much more than other compounds, absolutely not ‘compound 7 (too little)’ as recorded on Page 6.
A: I am very sorry for the mistake, the compound 8 (0.7 mg) was not subjected to cytotoxicity.
Question 4: ‘compound 5 not only inhibits HCT116 cells proliferation but also induces apoptosis…The mechanism of action on these two cell lines could ascribe to the large difference in their IC50 values.’ Was this consistent with that the cytotoxicity IC50 values of compound 5 against HCT116 (16.7 μM) was more than against CT26 (14.9 μM) as shown in Table 2?
A: Sorry, this sentence is ambiguous. I want to express compounds 3 and 16 have large difference IC50 values in HCT116 and CT26 cell lines. To avoid ambiguity, we deleted the entire sentence. (Page 7)
Corrections:
- Abstract: ‘Several, studies’ → ‘Several, studies’
‘16 known species’ → ‘16 known analogues’
‘Cytotoxic activity screening identified, three compounds cytotoxicity in a dose-dependent manner’ → ‘Cytotoxic activity screening revealed three compounds exhibited cytotoxicity in a dose-dependent manner’
- Introduction: ‘exert positive effects, re-duce fever, and prevent malaria’ → ‘exerting positive effects on reducing fever and preventing malaria’
‘Several, researchers’ → ‘Several researchers’
- Results and Discussion:
P2: ‘aerial part f A. annua’ → ‘aerial part of A. annua’
‘extracted in ethyl acetate (EtOAc) and n-butanol to obtain three residues purified by column chromatography’ → ‘extracted by ethyl acetate (EtOAc) and n-butanol to afford three residues, which were purified by column chromatography’
‘compounds 1–2’ → ‘compounds 1 and 2’
‘artemanins B’ → ‘artemanin B’
‘Arteannoide I’ → ‘arteannoide I’
‘Qinghaosu I’ → ‘qinghaosu I’
P6: The expression ‘a sodium adduct ion peak at m/z 249.1488 [M + H]+’ was wrong.
A:I am very sorry for the mistake. The sentence was changed to ‘Compound 2 was obtained as yellow oil, with the molecular formula of C15H20O3 based on its HRESIMS (m/z 249.1488 [M + H]+ , calculated for 249.1485), and its molecular formula was C15H20O3.’ (Page 6)
‘olefinic methane’ → ‘olefinic methine’
This sentence ‘except for the disappearance of the exomethylene (δH 4.86, 4.79, δC 113.0) in 1 and the presence of olefinic methane (δH 5.44, δC 121.6) in 2, as well as the position of a hydroxyl group’ was obscure.
A:The sentence was changed to ‘The major difference was an exomethylene group (δH 4.86, 4.79, δC 113.0) was present in 1 instead of the olefinic methane (δH 5.44, δC 121.6) in 2’. (Page 6)
‘experimental values of 5 using the DP4+’ → ‘experimental values of 2 using the DP4+’
‘its absolute configuration 1 was established’ → ‘its absolute configuration was established’
P6: ‘of inhibited’ → ‘of inhibiting’
References: Please check and revise some references according to the Instructions for Authors.
A: Thank you very much for your revisions to the sentences in the manuscript, we have made the revisions one by one as listed. The references have been revised according to the author's instructions.

Reviewer 2 Report
In this manuscript, Han et al. isolated 18 sesquiterpenes from the herb, including 2 new sesquiterpenes and 16 known species. For new synthesized structures were elucidated by comprehensive spectroscopic analyses, including HR-ESI-MS, NMR experiments, single-crystal X-ray, and DP4+ and electronic circular dichroism (ECD) calculations. Сytotoxic activity was also screened. However, compared to the classical CPT analogues, such results are unconvincing. Besides, considering the cost of raw materials, the convenience of one-step synthesis is debatable.
And yet despite the noted disadvantages, I think the manuscript might meet the requirements for publication in Molecules.
I do feel that some details should, however, be added or clarified.
1. On pages 1 of introduction section, the authors have discussions on the important role of Artemisia annua (Aa) as effect treatment. A quite related paper (Int. J. Mol. Sci. 2020, 21, 4986; Mini Reviews in Medicinal Chemistry, Volume 19, Number 11, 2019, pp. 902-912 (11) that involved various activities of (Aa) should be cited as the reference.
2. There is no conclusion in the work, which reduces the value of the work.
3. Although the two compounds are synthesized from the same raw materials, they seem to have no practical connection. The novelty and significance of this work needs to be further highlighted.
Author Response
Dear Reviewer,
Thank you very much for your advice on our manuscript titled “Sesquiterpenes from Artemisia annua and their cytotoxic activities (molecules-1845823)”. We submitted the revised manuscript as suggested. We have addressed the comments raised by reviewers, and the amendments are highlighted in red in the revised manuscript. We hope that the revision is acceptable and we look forward to hearing from you soon.
- On pages 1 of introduction section, the authors have discussions on the important role of Artemisia annua (Aa) as effect treatment. A quite related paper (Int. J. Mol. Sci. 2020, 21, 4986; Mini Reviews in Medicinal Chemistry, Volume 19, Number 11, 2019, pp. 902-912 (11) that involved various activities of (Aa) should be cited as the reference.
A: Thank you very much for your suggestion, these 2 references are very important, we added it in the second paragraph of Introduction. (Page 1, reference 9 and 10)
- There is no conclusion in the work, which reduces the value of the work.
A: Thank you very much for your suggestion, we added the conclusion as the third subsection. (Page 8, 3. Conclusion)
- Although the two compounds are synthesized from the same raw materials, they seem to have no practical connection. The novelty and significance of this work needs to be further highlighted
A: Thank you very much for your suggestion, we further discuss the importance and novelty of the work in the conclusion (Page 8), and the structures relationship of these compounds were discussed in the third paragraph of Introduction. (Page 2)
Round 2
Reviewer 1 Report
The manuscript has been clearly improved. No further suggestion from me. Now it is a nice paper, and I recommend it for publication.